# Sirtuin 1 in Host Defense during Infection

**DOI:** 10.3390/cells11182921

**Published:** 2022-09-19

**Authors:** Jin Kyung Kim, Prashanta Silwal, Eun-Kyeong Jo

**Affiliations:** 1Department of Microbiology, Keimyung University School of Medicine, Daegu 42601, Korea; 2Department of Microbiology, Chungnam National University School of Medicine, Daejeon 35015, Korea; 3Infection Control Convergence Research Center, Chungnam National University School of Medicine, Daejeon 35015, Korea

**Keywords:** sirtuin 1, infection, AMPK, bacteria, virus, parasite

## Abstract

Sirtuins (SIRTs) are members of the class III histone deacetylase family and epigenetically control multiple target genes to modulate diverse biological responses in cells. Among the SIRTs, SIRT1 is the most well-studied, with a role in the modulation of immune and inflammatory responses following infection. The functions of SIRT1 include orchestrating immune, inflammatory, metabolic, and autophagic responses, all of which are required in establishing and controlling host defenses during infection. In this review, we summarize recent information on the roles of SIRT1 and its regulatory mechanisms during bacterial, viral, and parasitic infections. We also discuss several SIRT1 modulators, as potential antimicrobial treatments. Understanding the function of SIRT1 in balancing immune homeostasis will contribute to the development of new therapeutics for the treatment of infection and inflammatory disease.

## 1. Introduction

Mammalian sirtuins (SIRTs) are a family of nicotinamide adenine dinucleotide (NAD^+^)-dependent deacetylases, homologous to yeast silent information regulator 2 (SIR2) [1]. Because SIRTs sense NAD^+^ fluctuations, their functions are indispensable in the control of energy and metabolic homeostasis as well as in redox regulation [2,3]. In addition, SIRTs are critical regulators of many cellular functions, including longevity, immune and inflammatory responses, DNA damage repair, unfolded protein response, and cell apoptosis [4,5,6]. So far, seven SIRTs (SIRT1–7), each with a distinct distribution and subcellular localization, have been reported in mammalian cells [4,5,7]. SIRTs are classified into four groups according to their sequence similarities, with class I comprising SIRT1, SIRT2, and SIRT3; class II, SIRT4; class III, SIRT5; and class IV, SIRT6 and SIRT7 [8]. Although SIRTs are specialized lysine deacetylases, all SIRTs except SIRT5 also possess ADP-ribosyl transferase activity [8]. SIRT5 removes succinyl, malonyl, and glutaryl groups from target proteins, mainly those involved in mitochondrial metabolism [7,9,10]. Among the SIRTs, SIRT1, which is critically involved in host defense against various infections, has been the most extensively studied [11,12,13,14].

In response to various infectious stimuli, host innate and adaptive immune defense systems are activated. The innate sensing of viruses, bacteria, fungi, and parasites results in the initiation of intracellular signaling pathways that activate the innate host defense system and therefore the inflammatory response, the release of antimicrobial mediators, and the intracellular killing of microorganisms [15,16,17]. Macrophages are myeloid lineage cells and in the innate immune response they are the first barrier against invading pathogens [18]. The most prominent feature of the adaptive immune system is immune memory, which allows a faster and amplified response to subsequent antigen challenge [19,20]. T and B lymphocytes play crucial roles in the integration of cellular and humoral immune responses, respectively [19,20,21]. The innate and adaptive immune systems are interconnected and their bidirectional interplay results in effective host defenses during infection.

Studies of SIRTs have clearly demonstrated the regulatory functions of these proteins in host defenses following infection, including epigenetic modulation of the target genes involved in immune and inflammatory responses, autophagy, and the immunometabolic remodeling of immune cells. However, the function(s) of each SIRT within immune cells and the multiple intracellular pathways that serve to integrate host defenses during infection are poorly understood. In the following, we provide an overview of the role of SIRT1 in modulating innate and adaptive immune responses, autophagy, and immunometabolism during bacterial, viral, and parasitic infection. Although the function and molecular mechanisms by which SIRT1 regulates the inflammatory response during sepsis have been extensively reviewed elsewhere [22], our focus is on the role of SIRT1 in the regulation of host defenses against infection.

## 2. Overview of SIRTs

All SIRT family members play distinct but also overlapping roles in the regulation of physiological responses, depending on the nature of the trigger and the cell type. Seven SIRTs have been described so far. SIRT1 is a nutrient/metabolic sensor [23] that is mainly localized in the nucleus but able to shuttle between the nucleus and cytoplasm under certain conditions [24,25]. It targets a wide range of cellular proteins for post-translational modification by deacetylation [26], including multiple histone and non-histone proteins. In addition to its role in numerous important cellular and physiological processes, such as gene transcription, cellular senescence, aging, tumorigenesis, mitochondrial function, metabolism, oxidative stress, the cell cycle, circadian rhythm, and autophagy, SIRT1 plays a key role in the regulation of inflammation and immune function [1,27,28,29,30,31,32,33,34].

SIRT2, mainly located in the cytosol but able to translocate into the nucleus, regulates mitosis, cell differentiation, mitophagy, and cardiac homeostasis [35,36,37]. Mitochondrial SIRTs, including SIRT3, SIRT4, and SIRT5, are located in the mitochondrial matrix and are critically involved in the regulation of mitochondrial stress responses, antioxidant responses, apoptosis, and autophagy [38,39]. SIRT6 and SIRT7 are nuclear proteins, with SIRT7 primarily found in the nucleolus [40]. Nuclear SIRTs, including SIRT1, 2, 6, and 7, act as regulators of the inflammatory response [41]. Given the essential function of SIRTs in numerous biological pathways, their dysregulation and abnormal expression are, not surprisingly, associated with a variety of human diseases, including metabolic and neurodegenerative diseases, cardiovascular diseases, cancers, infection, and inflammatory diseases [42,43,44,45]. There is also a large body of evidence demonstrating a role for SIRTs in the epigenetic control of several target genes and proteins, by the modification of histone and non-histone proteins in both normal and transformed cells. Considering the role of SIRTs in the regulation of vital physiological functions, their further investigation will contribute to the development of new therapeutic agents for the treatment of human diseases, including infections.

The development of improved therapeutics against various infections will depend on a better understanding of the key factor(s) and the mechanisms underlying the regulation of host–pathogen interactions during infections. SIRT1, one of the most well-studied sirtuins, has become recognized as a critical modulator in the diverse aspects of host immune and inflammatory responses in the context of infection. SIRT1 regulates autophagy, a cell-autonomous host defense pathway, by the NAD-dependent deacetylation of autophagy-related proteins [33,46]. Upon oxidative stresses, SIRT1 can activate autophagy and mitochondrial function through suppression of the mammalian target of rapamycin (mTOR) pathway in embryonic stem cells [47]. In addition, SIRT1 regulates the expression of hypoxia-inducible factor 1α (HIF-1α) and vascular endothelial growth factor to modulate airway inflammation [48]. In contrast, SIRT1, through activating mTOR, contributes to hepatic inflammation, inflammasome activation, and liver injury [49]. Future studies will provide insights into the SIRT1 crosstalks to mTOR pathway to alleviate discrepancies in different experimental settings.

A growing body of literature suggests an essential role for SIRT1 in regulating immunometabolic pathways to impact the immune and inflammatory responses [49,50]. Upon TLR4 stimulation, SIRT1, coupling to a cofactor NAD^+^, regulates epigenetic reprogramming for the coordinated bioenergetic and inflammatory responses [51,52]. In addition, the SIRT1/AMP-activated protein kinase (AMPK) axis contributes to NAD^+^-induced amelioration of experimental autoimmune encephalomyelitis through modulating Th1/Th17 immune responses [53]. Moreover, SIRT1-deficient dendritic cells drive Th1 immune responses and microbial inflammation [54], suggesting that SIRT1 is involved in T cell differentiation during infection. It is also well known that AMPK is critical for the energy generation mechanisms such as the autophagy pathway [55]. Indeed, numerous studies have shown the function of SIRT1/AMPK axis in the autophagy regulation [56,57,58], although it remains largely unknown in the context of infections. During respiratory syncytial virus (RSV) infection, SIRT1 in dendritic cells is required for the appropriate immunometabolic responses through AMPK-acetyl CoA carboxylase (ACC) pathway via controlling catabolic and anabolic reactions [59]. Among the downstream factors deacetylated by SIRT1 in the regulation of transcriptional and metabolic pathways are NF-κB and HIF-1α, both of which are critically required for immunometabolic and inflammatory functions [54,60]. Given the prominent role of AMPK in the activation of autophagy, SIRT1 appears to be a crucial orchestrator of autophagy and immunometabolism, and both pathways are interconnected through mTOR/AMPK axis [50,61,62], thereby impacting innate and adaptive immune responses in response to several infectious and inflammatory stimuli (Figure 1). Below, we discuss more detailed roles of SIRT1 in the past and recent studies of various infections.

## 3. Function of SIRT1 during Infections

Although SIRTs mediate a variety of biological functions, leading to diverse cellular responses [7,63,64], their role in the regulation of innate immune responses to bacterial and viral infections has yet to be fully addressed. SIRT1 is the most widely studied SIRT family member in terms of the host defense and immune regulation against various infections. Thus, in the following we review what is known about SIRT1 in terms of its regulation of host immune responses during bacterial and viral infections.

### 3.1. SIRT1 in Bacterial Infections

Although SIRT1 was initially shown to have little influence on the regulation of mortality in Gram-negative endotoxemia or Gram-positive bacterial infection [65], a subsequent study demonstrated its involvement in models of several bacterial infections, including *Helicobacter pylori* infection. In the latter, SIRT1 is downregulated in gastric cells such that its suppression of the bacterium is inhibited [13]. The transcriptional factor RUNX3 is responsible for SIRT1 expression in gastric cells [13]. Mechanistically, SIRT1 improves the autophagic flux, which is defective in *H. pylori*-infected cells [13], and is thus a promising target for the eradication of *H. pylori* infection. Furthermore, the *Salmonella* virulence factor pathogenicity island 2 enables the evasion of host autophagy, by degrading the SIRT1/LKB1/AMPK complex and by the sustained activation of the mTOR pathway [66]. These data suggest that SIRT1-mediated antibacterial autophagy contributes to host defense activities, although the molecular mechanisms are largely unknown.

In epithelial cells infected with *Streptococcus pneumoniae*, SIRT1 activation by resveratrol promotes, whereas SIRT1 inhibition by nicotinamide suppresses, the expression of the human β-defensin gene [11]. However, whether SIRT1 is involved in the elimination of *S. pneumoniae* in host cells is unclear. During infection with Gram-positive *Staphylococcus aureus*, SIRT1-mediated regulation of IL-9 was shown to modulate inflammatory responses induced by the bacterium [67]. IL-9 inhibition activates anti-inflammatory responses in mice infected with methicillin-resistant *S. aureus* [67]. A recent study showed that miR-155 targets SIRT1 to upregulate IL-9 levels and Th9 cells [68]. These findings suggest that SIRT1 activation enables the modulation of inflammatory responses potentially harmful to the host during Gram-positive bacterial challenge.

Tuberculosis (TB) is a serious infectious disease caused by *Mycobacterium tuberculosis* (Mtb). Several studies have shown that SIRT1 is critically required for anti-mycobacterial responses [46,68,69]. SIRT1 activation suppresses the intracellular growth of Mtb, including that of drug-resistant strains, and promotes both autophagy and phagosome-lysosome fusion. In addition, beneficial effects on pulmonary pathology, chronic inflammation, and the efficacy of anti-TB drugs were demonstrated in mice treated with SIRT1 activators [69]. The interaction of SIRT1 with the orphan nuclear receptor estrogen-related receptor-α (ERRα) is critically required in the antimicrobial host defense against Mtb infection [46]. Indeed, a positive feedforward loop between ERRα and SIRT1 was shown to contribute to the activation of antibacterial autophagy against Mtb infection, through the deacetylation of essential autophagy proteins, including autophagy-related gene (ATG)5, BECN1, and ATG7 [46]. Yang et al. showed that the interaction of SIRT1 with TAK1 inhibits the activity and downstream signaling of TAK1, resulting in the suppression of proinflammatory cytokine generation [14]. Mice treated with the SIRT1 agonist resveratrol are more resistant to Mtb infection, indicating the potent therapeutic effects of the drug in the innate defense against Mtb [14]. Together, these studies demonstrate the potential of SIRT1 activators in host-directed therapeutics against human TB.

### 3.2. SIRT1 in Viral Infections

Over the last decade, much has been learned about the function of SIRT1 in the regulation of host defenses against viral infection. Human immunodeficiency virus (HIV) type 1 Tat protein suppresses SIRT1, thereby blocking the deacetylation of NF-κB and the hyperactivation of T cells, which are two common features of the chronic phase of HIV infection [70]. Indeed, SIRT1 mRNA levels are low in the peripheral blood mononuclear cells (PBMCs) of treated HIV patients [71]. Another study showed that the downregulation of SIRT1 is linked to the upregulation of Period circadian clock 2 (Per2) and may contribute to the diminished expansion of hematopoietic progenitor cells (HPCs) in patients with chronic HIV infection [72]. The SIRT1 activator resveratrol induces SIRT1 mRNA in PBMCs and decreases Per2-expressing HPCs in HIV+ patients [72]. During combined antiretroviral treatment in HIV infection, the plasma SIRT1 level is higher in patients treated with integrase transfer inhibitors than in to those treated with protease inhibitors [73]. These data suggest a significantly diminished SIRT1 level in HIV patients and that SIRT1 is differentially regulated depending on the treatment regimen. However, the mechanism by which SIRT1 levels are modulated by HIV infection or treatment remains to be determined.

SIRT1 may play a role in antiviral responses against influenza virus infection by modulating cell senescence. A previous study showed that the SIRT1 inhibitor nicotinamide and the class I HDAC inhibitor sodium butyrate (NaB) increase viral plaques and replication [74]. In hepatitis C virus (HCV) infection, manipulation of the SIRT1 pathway leads to the control of HCV-mediated T cell senescence and increased IL-2 production in senescent CD4^+^ T cells, at least partly, through a reduction of regulatory T cells [75]. During HCV infection, the transcription factor ΔNp63-mR-181a axis is critically required for SIRT1 activity. The decline of miR-181a upregulates SIRT1, which restores impaired T cell responses, by functionally activating CD4^+^ T cells during chronic HCV infection [75]. Some studies have strongly suggested the rejuvenating function of SIRT1 in terms of immune senescence, but others have reached opposite conclusions. A recent study suggested a role for SIRT1 in the susceptibility of older adults to viral infections [76]. Interestingly, in this population, the suppression of SIRT1 activity increases histone expression and cell cycle progression in replicating T cells, thereby restoring the replication-stress response. In addition, SIRT1 inhibitors enhance viral clearance in mouse models of lymphocytic choriomeningitis virus (LCMV) infection [76]. The age-associated decrease in the expression of miR-181a in T cells has been linked to the miR-181a targeting activity of SIRT1, implying a role for SIRT1 in cell replication stress and delayed viral clearance during infection in older adults [76]. During RSV infection, SIRT1 inhibition by the selective inhibitor EX-527 exacerbates the pathological changes in the lung and elevates the viral load in vivo [77]. Thus, SIRT1 function and activity are complex, presumably due to different age/study populations, clinical stages, and experimental models.

The role of SIRT1-mediated autophagy in the antiviral responses during RSV infection was also investigated. Dendritic cell (DC)-specific SIRT1 knockout mice have an exacerbated lung pathology during RSV infection [77]. These data suggest the beneficial function of SIRT1 in the promotion of antiviral immune responses through the DC-mediated activation of autophagy [77]. In response to RSV infection, higher levels of interleukin-*1β*, -*6*, and -*23* are induced at early time points by SIRT1-deficient bone marrow DC (BMDCs) than by wild-type BMDCs. SIRT1 deficiency in DC results in defective mitochondria and increased levels of proinflammatory cytokines that prime Th17 responses and thus amplify RSV-induced inflammation [59]. The same study proposed that the SIRT1-mediated ACC inhibition that regulates fatty acid synthesis and pathological inflammation is required for effective antiviral immunity [59]. Together, the SIRT1 through autophagy activation could serve as a therapeutic target against RSV-induced inflammation.

There is increasing evidence of SIRT1 involvement in the pathogenesis of coronavirus disease-2019 (COVID-19), the causative agent of the recent pandemic. In COVID-19 patients, p53 and proinflammatory cytokine levels are increased, but SIRT1 expression in PBMCs is decreased [78]. A recent study reported that SIRT1 is involved in the inhibitory action of (d-Ala^2^)-dynorphin 1-6 (leytragin), a peptide agonist of δ-opioid receptors, upon high mobility group box 1 (HMGB1) secretion in the lungs of mice with lipopolysaccharide (LPS)-induced acute lung injury [79]. Because extracellular HMGB1 is strongly associated with the clinical severity of COVID-19, leytragin-induced SIRT1 activation may contribute to the development of HMGB1-targeting therapeutics and thus to the prevention of the cytokine storms that often accompany COVID-19 [79]. Further research into the potential of SIRT1 activators in the amelioration of excessive inflammatory responses may provide a treatment strategy for the acute lung injury and cytokine storms often associated with COVID-19 pathology.

### 3.3. SIRT1 in Parasitic Infections

Because parasitic protozoa and helminths also have SIRTs, here we focus on the host SIRT1. Several findings suggest that SIRT1 participates in the improvement of host defenses against protozoal infection. In chronic Chagas disease, caused by *Trypanosoma cruzi*, mitochondrial-dysfunction-mediated superoxide release is associated with myocardial pathologies [80]. SIRT1 may play a beneficial role in the amelioration of the sustained oxidative stress and disease pathology resulting from *T. cruzi* infection [80,81]. For example, the SIRT1 agonist SRT1720 suppresses proliferation and proinflammatory responses in *T. cruzi*-infected macrophages, thus reducing the chronic inflammation that occurs in Chagas disease [82]. Malaria is a life-threatening infection caused by the genus *Plasmodium*. A recent study using a mouse model of cerebral malaria showed that hypothyroidism promotes mouse survival, with protection attenuated by the SIRT1 inhibitor EX-527, whereas in mice with normal thyroid function the SIRT1 activator SRT1720 confers host protection [83], thus suggesting SIRT1 as a therapeutic target in cerebral malaria.

However, an opposing role for SIRT1 in anti-parasitic host responses has also been described. In leishmaniasis, caused by *Leishmania infantum*, the metabolic sensor SIRT1 or AMPK was shown to be detrimental to host defenses. A deficiency of SIRT1 or LKB1 suppressed the parasite loads of macrophages in vivo and inhibited the expression of genes required for parasite growth, such as Slc2a4 [84]. Infection with the parasite *Leishmania donovani* triggers SIRT1 induction to inactivate forkhead box O (FOXO)-1 through deacetylation, thus favoring parasite survival in host cells [85]. Either silencing SIRT1 or use of the SIRT1 inhibitor sirtinol significantly reduced the parasite burden and upregulated the production of tumor necrosis factor α, reactive oxygen species (ROS), and nitric oxide [85]. Therefore, therapeutic approaches against parasitic infections await a better understanding of the complex interactions between parasites and the host SIRT1 network.

Following infection with *Cryptosporidium parvum*, a zoonotic protozoan parasite, the expression of let-7i decreased, but SIRT1 levels in human biliary epithelial cells increased. *C. parvum*-mediated upregulates SIRT1 functions in the modulation of NF-κB activation and in epithelial cell innate responses [86]. However, the exact function of SIRT1 in *C. parvum* clearance by host epithelial cells is unclear (Table 1). Further studies of the mechanisms by which parasites exploit and/or modulate SIRT1-related host responses will improve our understanding of the precise role of SIRT1 in parasitic infections and contribute to the development of innovative anti-parasitic therapeutics.

## 4. Conclusions

The cellular stresses caused by infectious pathogens are closely associated with the depletion of NAD^+^, a key coenzyme required for the bioenergetics of cells. Accumulating evidence suggests that SIRT1, a widely-studied member of SIRT NAD^+^-dependent deacetylases, plays a key role in shaping antimicrobial host defenses, by modulating inflammation, immunometabolism, and ROS generation. Recent studies have provided detailed descriptions of SIRT1′s functions in bacterial, viral, and parasitic infections, but less is known about the protein’s role in host defenses against fungal infection. In addition, most studies have reported the SIRT1 regulation of innate and inflammatory responses in the context of infection; future studies are needed to clarify the exact function of SIRT1 in regulating adaptive immune responses to impact host defense. Moreover, host regulation of SIRT1 in response to each pathogen and the mechanisms by which pathogens modulate SIRT1 to evade host antimicrobial responses are largely unknown. In view of the vital physiological function of SIRT1 in the regulation of mitochondrial metabolism, future research should be focused on the molecular mechanisms by which SIRT1 functions in the immunometabolic remodeling required for host defenses during infection. Finally, the possibility of using SIRT1 agonists or inhibitors in the control of infections should be investigated in experimental and clinical studies. These efforts may contribute to SIRT1-related interventions in acute and chronic infectious diseases and/or to combined therapy with currently-used antibiotics.

## Figures and Tables

**Figure 1 cells-11-02921-f001:**
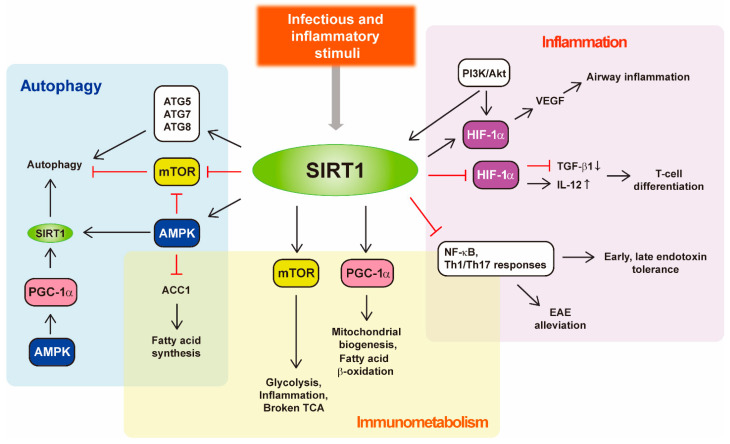
SIRT1 regulates autophagy, inflammation, and metabolic reprogramming. SIRT1 directly deacetylates components of the autophagy (ATG5, ATG7, and ATG8) and inhibits the mTOR pathway to activate autophagy. The activation of AMPK-PGC-1α pathway induces the expression of SIRT1 and promotes autophagy activation. In addition, SIRT1-mediated deacetylation of AMPK inactivates ACC1 so that it decreases fatty acid synthesis during infection. The SIRT1/mTOR axis contributes to increased glycolysis, a disrupted TCA cycle, and altered inflammasomes in response to endotoxin. SIRT1 controls mitochondrial biogenesis and fatty acid β-oxidation by activating PGC-1α. Additionally, SIRT1 regulates NF-κB signaling and pro-inflammatory T cell responses to alleviate EAE. SIRT1 is required for the endotoxin tolerance by deacetylating NF-κB during LPS stimulation. In dendritic cells, SIRT1 participates in T-cell differentiation, by modulating TGF-β1 and IL-12 production. In a murine model of allergy induced by ovalbumin-inhalation, SIRT1 regulates VEGF expression through the HIF-1α, thus controlling airway inflammation. Abbreviations: ACC, acetyl CoA carboxylase; ATG, autophagy-related gene; EAE, encephalomyelitis; PI3K, phosphoinositide 3-kinase; TCA, tricarboxylic acid cycle; VEGF, vascular endothelial growth factor.

**Table 1 cells-11-02921-t001:** Roles of SIRT1 during bacterial, viral, and parasitic infections.

Pathogen	Cells/Study Model	SIRT1 after Infection	Intervention	Outcome	Mechanism of Action	Ref.
**Bacterial Infection**						
*Helicobacter pylori*	Human gastric cell lines, gastric tissues	↓	RUNX3 knockdown or overexpression, SRT1720	Increased intracellular survival and colonization	Inhibition of SIRT1 in a RUNX3-dependent manner, and inhibition of autophagy flux	[13]
*Salmonella* Typhimurium	Murine BMDMs	↓	Sirt1 overexpression, Torin 1, Atg7^−/−^	Compromised autophagic host cell defense	AKT-mTOR-dependent degradation of SIRT1 and inhibition of AMPK activation and autophagy by virulence factor SPI2	[66]
*Streptococcus pneumoniae*	A549 cell line	↑	Resveratrol, Nicotinamide	Induction of hBD2 and IL-8	SIRT1-mediated induction of hBD2 through p38 MAPK and IL-8 through ERK	[11]
*Staphylococcus aureus*	BALF from children with MRSA, CD4^+^ T cells	↓ *	miR-155 mimic and inhibitor, SIRT1 overexpression	Th9 differentiation	miR-155-mediated negative regulation of SIRT1	[68]
*Mycobacterium tuberculosis*	THP-1 cells, hMDMs, in vitro mouse model	↓	SRT1720, SIRT1 KO	Higher TB pathogenesis	SIRT1 activation-mediated induction of autophagy and phagosome-lysosome fusion, and deacetylation of RelA/p65	[69]
	HEK293T cells, human PBMCs, Mouse PMs	↓	TLR2^−/−^, Sirt1^−/−^, Resveratrol	Negative regulation of inflammatory responses	Enhanced TAK1 phosphorylation/ubiquitination leading to activation of p65/MAPK signaling pathway	[14]
	BMDMs, RAW264.7 cells, HEK293T cells	Not reported	AICAR, Resveratrol, XCT790, esrra^−/−^	Antibacterial autophagy	AMPK or SIRT1 activation leading to induction of ESRRA	[46]
**Viral Infection**						
Human immunodeficiency virus (HIV)	Jurkat T cells, HeLa, HEK293, MEF cells	↓	SIRT1^−/−^, Tat overexpression, Nicotinamide	Immune cell hyperactivation	Interaction of Tat with SIRT1 to block its deacetylase activity and superinduction of T cell activation and HIV transcription via NF-kB	[70]
	PBMCs from HIV+ and healthy individuals	↓	Resveratrol	Impaired immune reconstitution and accelerated aging process	Negative regulation of Per2 by Sirt1 in HPCs	[72]
Hepatitis C virus (HCV)	PBMCs from healthy and HCV infected patients	↑	SIRT1 and ΔNp63 knockdown, miR-181a precursor	T cell senescence and viral persistence	Upregulation of ΔNp63 by HCV leading to inhibition of miR-181a, thereby increasing Sirt1 and DUSP6 in CD4^+^ T cells	[75]
Lymphocytic choriomeningitis virus (LCMV)	Human primary T cells	Not reported	EX-527, SIRT1 knockdown, miR-181a^−/−^	Diminished cell cycle progression and replication stress	Reduction of histone gene by miR-181a-dependent increase of SIRT1	[76]
Respiratory syncytial virus (RSV)	mouse BMDCs, AECs, in vivo mouse model	↑	EX-527, *Sirt1* KO, SIRT1 knockdown, SRT1720	Antiviral immune response	Induction of autophagy, induction of Th1 immune response, suppression of Th2 and Th17 responses	[77]
	BMDCs, in vivo mouse model	Not reported	SIRT^−/−^, C75	Altered immune homeostasis	In SIRT1-deficient DCs—reduction of mitochondrial function/respiration andincreased fatty acid synthesis through acetyl Co A pathway	[59]
COVID-19	PBMCs from COVID-19 patients	↓	-	Altered lymphocyte homeostasis	Increased p53 and reduced SIRT1, reduced expression of IL17R and BLNK	[78]
**Parasitic Infection**						
*Trypanosoma cruzi*	Chagas mice, RAW264.7 cells	Not reported	SRT1720, PF-562271 (iFAK)	Inhibition of inflammation during Chagas disease	Suppression of FAK phosphorylation, FAK-dependent increase of Pu.1 and Runx1, and NF-κB activity	[82]
*Plasmodium berghei* ANKA (PbA)	ECM mice	↓	SRT1720, EX-527, Hypothyroid mice	Improved disease outcome	Sirt1 as a mediator of the action of the thyroid hormones	[83]
*Leishmania infantum*	BMDMs, PMs, in vivo mouse model	↓	SIRT1 KO, LKB1 KO, SRT1720	Parasitic growth	Macrophages switch from early glycolytic metabolism to oxidative phosphorylation through SIRT1 and LKB1/AMPK	[84]
*Leishmania donovani*	RAW264.7 cells, In vivo mouse model	↑	Sirtinol, AS1842856, SIRT1 knockdown	Parasitic survival	Deacetylation and inactivation of FOXO-1 by SIRT1 to prevent apoptosis	[85]
*Cryptosporidium parvum*	H69 cells	↑	Pre-let-7i, Anti-let-7i	-	Reduction in let-7i leading to inhibition of miRNA-mediated translational suppression of SIRT1 to induce its protein level	[86]

Abbreviations: AEC, alveolar epithelial cell; Akt, protein kinase B; AMPK, AMP-activated protein kinase; Atg7, autophagy-related gene *7*; BALF, bronchoalveolar lavage fluid; BD2, beta-defensin 2; BLNK, B Cell linker; BMDC, bone marrow dendritic cell; BMDM, bone marrow-derived macrophage; DC, dendritic cell; DUSP6, dual specificity phosphatase 6; ECM, experimental CM; ESRRA, estrogen related receptor alpha; FAK, focal adhesion kinase; FOXO-1, forkhead box O-1; HPCs, hematopoietic progenitor cells; IL17R, interleukin 17 receptor; KO, knock out; MDM, monocyte-derived macrophage; MEF, mouse embryonic fibroblasts; MRSA, methicillin-resistant *Staphylococcus aureus*; mTOR, mammalian target of rapamycin; PBMC, peripheral blood mononuclear cell; Per2, period circadian clock 2; PM, peritoneal macrophage; SPI2, pathogenicity island 2; TAK1, transforming growth factor-β-activated kinase 1; TB, tuberculosis; TLR2, toll-like receptor 2. * (Not direct but miR-155 mimic inhibits SIRT1) MRSA → high miR-155; miR-155 mimic → Sirt1 inhibition.

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
