# Peer review of "Sirtuin 1 in Host Defense during Infection"

_cells, 2022, doi:10.3390/cells11182921_

Round 1
Reviewer 1 Report
This review is well structured and organized.
Few considerations:
1- taking into consideration that this review is focused on SIRT1 action during infection, what about the role of SIRT1 in the adaptive immune system?
2- It is not clear the function od SIRT1 in the regulation on crosslink between autophagy and immunometabolism
3- figure 1 is not easy to understand. All indicated “inductors” (H2O2, MOG, RVS… ) are able to activate SIRT1 and give the same cellular response?
4-what about SIRT1 role in fungal infections?
5- please correct some mistakes in writing style (e.g in paragraph 2)
Author Response
1- taking into consideration that this review is focused on SIRT1 action during infection, what about the role of SIRT1 in the adaptive immune system?
Responses:
Thanks for your point. As we commented in the overview, the SIRT1/AMP-activated protein kinase (AMPK) axis contributes to NAD+-induced amelioration of experimental autoimmune encephalomyelitis through modulating Th1/Th17 immune responses [53]. However, in the context of various infectious diseases, the SIRT1 function in regulation of adaptive immune responses have not been fully understood. In addition, there is a word limitation. So we just briefly mention about this issue as perspectives in the conclusion to facilitate future studies in the field.
In addition, most studies have reported the SIRT1 regulation of innate and inflammatory responses in the context of infection; future studies are needed to clarify the exact function of SIRT1 in regulating adaptive immune responses to impact host defense.
2- It is not clear the function od SIRT1 in the regulation on crosslink between autophagy and immunometabolism
Responses:
Although it is an important aspect to discuss autophagy and immunometabolism, the review suffers from the word constrain. So we just briefly mention about this issue in the introduction, as follows:
Given the prominent role of AMPK in the activation of autophagy, SIRT1 appears to be a crucial orchestrator of autophagy and immunometabolism, and both pathways are inter-connected through mTOR/AMPK axis [50,61,62], thereby impacting innate and adaptive immune responses in response to several infectious and inflammatory stimuli (Figure 1).
3- figure 1 is not easy to understand. All indicated “inductors” (H2O2, MOG, RVS… ) are able to activate SIRT1 and give the same cellular response?
Responses:
We have corrected various ligands as ‘Infectious and inflammatory stimuli’. The detailed information has been included in the text.
4-what about SIRT1 role in fungal infections?
Responses:
As we mentioned in the conclusion, we could not find the original articles about the function of SIRT1 in the context of fungal infection. So we just discuss that future studies are needed in fungal infections.
Recent studies have provided detailed descriptions of SIRT1’s functions in bacterial, viral, and parasitic infections but less is known about the protein’s role in host defenses against fungal infection.
5- please correct some mistakes in writing style (e.g in paragraph 2)
Responses:
We have corrected the mistakes in writing style.
Reviewer 2 Report
Sirtuins are a family of NAD+ dependent deacetylases and have different regulatory functions in the immune and metabolic response, especially the innate immune responses to bacterial and viral infections. The authors carry out a review of SIRT1 of the 7 known, because the function of SIRT1 and its modulation during the infectious process is better known. The authors highlight that the members of the SIRT family are regulated from the triggers for the cellular response, although the authors recognized the participation of other SIRTs in different mechanisms such as the antioxidant response. They also refer to some experimental studies such as the agonist effect of SIRT1 that are more resistant to A Mtb infection. The authors describe the main ones identified to modulate bacterial, viral and fungal infections. The authors describe the role of SIRT in viral diseases such as HIV and other less frequent ones. They also describe the issue of Parasitic infections On the second sheet, last paragraph, the phrase “mammalian target or rapamysin” has a different font.
Author Response
Sirtuins are a family of NAD+ dependent deacetylases and have different regulatory functions in the immune and metabolic response, especially the innate immune responses to bacterial and viral infections. The authors carry out a review of SIRT1 of the 7 known, because the function of SIRT1 and its modulation during the infectious process is better known. The authors highlight that the members of the SIRT family are regulated from the triggers for the cellular response, although the authors recognized the participation of other SIRTs in different mechanisms such as the antioxidant response. They also refer to some experimental studies such as the agonist effect of SIRT1 that are more resistant to A Mtb infection. The authors describe the main ones identified to modulate bacterial, viral and fungal infections. The authors describe the role of SIRT in viral diseases such as HIV and other less frequent ones. They also describe the issue of Parasitic infections On the second sheet, last paragraph, the phrase “mammalian target or rapamysin” has a different font.
Responses:
Many thanks for your kind summary and comments. We have corrected the font of mTOR.